# Effect of a dietary intervention including minimal and unprocessed foods, high in natural saturated fats, on the lipid profile of children, pooled evidence from randomized controlled trials and a cohort study

**Rosanne Barbra Hendriksen**[1]*, **Ellen José van der Gaag**[2]

**1** MSc Nutrition and Health, Wageningen University and Research (WUR), Wageningen, The Netherlands,
**2** Department of Pediatrics, Hospital Group Twente (ZGT), Hengelo, The Netherlands

* rosannebh@gmail.com

## Abstract

### Aim

To study the possible effects of a dietary intervention with minimal and unprocessed foods, high in natural saturated fats on the lipid profile and body mass index of children.

### Method

This study combines three intervention studies; one non-randomized retrospective cohort study and two randomized controlled trials, to a pooled analysis. The intervention group received a dietary intervention of minimal and unprocessed foods for three to six months, consisting of five times per week green vegetables, three times per week beef, daily 200–300 mL whole cow's milk (3.4% fat) and whole dairy butter (80% fat) on each slice of bread. The control group continued their usual dietary habits. Raw data of the three intervention studies where combined into one single dataset for data analysis, using mixed effects analysis of covariance to test the effects of the dietary advice on the main study outcomes, which are measurements of the lipid profile.

### Results

In total, 267 children aged 1 to 16 years were followed. 135 children were included in the intervention group and 139 children in the control group. Characteristics (age, gender and follow-up period) were equally distributed between the groups at baseline. In the intervention group HDL-cholesterol increased significantly from 1.22 mmol/L, 95% confidence interval (CI) 1.14–1.32 to 1.42 mmol/L 95% CI 1.30–1.65 (p = 0.007). The increase over time in HDL cholesterol in the intervention group was significantly different compared to the increase in the control group (from 1.26 mmol/L, 95% CI 1.19–1.35, to 1.30 mmol/L, 95% CI 1.26–1.37) (p = 0.04). Due to the increased HDL concentration in the intervention group, the total

**Data Availability Statement:** All relevant data are within the manuscript and its Supporting information files.

**Funding:** The authors received no specific funding for this work.

**Competing interests:** The authors have declared that no competing interests exist.

cholesterol/HDL cholesterol ratio decreased significantly from 3.70 mmol/L, 95% CI 3.38–3.87, to 3.25 mmol/L, 95% CI 2.96–3.31 (p = 0.05).

## Conclusion

Consumption of minimal and unprocessed foods (high in natural saturated fats) has favourable effects on HDL cholesterol in children. Therefore, this dietary advice can safely be recommended to children.

## 1. Introduction

Worldwide the prevalence of childhood obesity is increasing dramatically. Childhood obesity often results in obesity in adults with its well-known negative effects on human health, like cardiovascular disease (CVD) [1]. Shockingly, overweight and obesity are linked to more deaths worldwide compared to underweight. According to the World Health Organization (WHO), in 2019, 38.2 million children worldwide under the age of 5 years were overweight or obese [2]. Changes must be made to fight this pandemic. The WHO suggests healthier food choices since these are the most accessible, available and affordable. However, UNICEF investigated so-called healthy food products specially made for children, such as porridge, breakfast cereals and snacks, and concluded that 70% of these 'healthy' products are actually unhealthy since they are high sources of energy, trans-fatty acids, sugar and sodium [3]. Many of these products are processed or ultra-processed. Several studies have shown that diets composed of (ultra-)processed food products are associated with negative health effects. Cornwell et al. (2018) obtained information from a cohort study conducted in children (aged 5–12 years) and observed that consumption of ultra-processed foods resulted in lower-quality nutrient profiles of the children [4]. Secondly, Rauber et al. (2015) examined the effects of ultra-processed food consumption in a longitudinal randomized trial, on 345 children's (aged 3–4 years and 7–8 years) lipid profiles [5]. It was concluded that early ultra-processed food consumption (at the age of 3–4 years) played a role in a negative altered lipid profile of the children later in life (at the age of 7–8 years). Eating more natural food products (unprocessed or minimally processed) at a younger age could contribute to a more positive lipid profile, and therefore to a healthier life.

Three previous conducted studies observed the effects of a dietary intervention consisting of minimal and unprocessed food products on children's lipid profiles [6–8]. This dietary intervention consisted of five times per week green vegetables at dinner, three times per week beef at dinner, daily 200–300 mL whole cow's milk and whole dairy butter on each slice of bread. There were some concerns about negative effects on the lipid profile of the young children that consumed these products since this dietary intervention was relatively high in saturated fats. However, there is conflicting research regarding the association of natural saturated fat intake and a negative lipid profile. A meta-analysis of prospective cohort studies did not find significant evidence that dietary saturated fat is associated with an increased risk of CVD [9]. Some studies even observed an inverse association. Mozaffarian et al. (2004) observed that a higher natural saturated fat intake led to a more favorable lipid profile with significant higher HDL cholesterol and lower total cholesterol/HDL cholesterol ratio in a randomized trial in postmenopausal women [10]. Secondly, Gillman et al. (1997) examined the association of stroke incidence with intake of fat and type of fat among middle-aged men from the US during 20 years of follow-up and concluded that natural fat, saturated fat, and monounsaturated fat

intake were associated with a reduced risk of stroke [11]. However, little research has been done on children.

Another point of discussion concerns the sustainability of this dietary advice. In 2019, the LANCET published the EAT-Lancet sustainable diet for the future. 37 scientists, from various disciplines, set global scientific goals for healthy nutrition and sustainable diets for the future [12]. Table 1 shows the comparison between the EAT-Lancet diet and the diet from the present study. The quantities of the LANCET diet are designed for adults [12]. No separate advice has been issued for younger children since they represent only a small part of the world population according to the authors of the EAT-Lancet committee. However, the dietary advice of the present study, designed for children, does fit within the guidelines of the EAT-Lancet diet. This dietary advice has shown in practice that it has clinical health-promoting effects. This illustrates that a healthy (unprocessed) diet within the global scientific goals for healthy nutrition and sustainable diets for the future, set by the EAT-Lancet committee, is feasible [6–8].

To investigate possible effects of this dietary intervention, a pooled analysis was performed using three previous conducted studies. The aim of the present study was to determine whether this dietary intervention, relatively high in saturated fats, has an influence on the lipid profile or body mass index (BMI) of young children.

## 2. Material and methods

The present study is a pooled analysis, combining three previous conducted studies all performed at ZGT in the Netherlands (Fig 1). The first study, a non-randomized retrospective cohort study, included children between 1 to 16 years, with at least two measurements of their lipid profile in the period between June 2011 and November 2013 at ZGT (n = 121) [6]. The aim of this retrospective cohort study was to determine whether the developed dietary advice had an influence on risk factors of cardiovascular disease in children. The second study, a randomized controlled trial, included children between 1 and 4 years old (n = 125), referred by their general physician with recurrent upper respiratory tract infections (URTIs), between March 2015 and October 2017, if they had a minimum of 3 URTIs in the last 3 months [8]. The aim of this randomized controlled trial was to evaluated whether the dietary advice can decrease the number and duration of URTIs in children with recurrent URTIs. The third study, a randomized controlled trial, included children aged 1 to 12 years (n = 65) in the period between January 2016 and September 2018, with subclinical hypothyroidism (SH) (Thyroid Stimulating hormone (TSH) > 4.2 mU/L and FT4 within the normal range) [7]. Here the aim was to investigate whether this dietary advice improves or normalizes SH or decreases the presence of Thyroid Stimulating Hormone (TSH) and/or tiredness.

In the three studies, participants from the intervention group received the same dietary advice, containing minimal and unprocessed food products. This intervention consisted of five times per week green vegetables, three times per week beef, daily 200–300 mL whole cow's

**Table 1. Components of the dietary advice of the present study compared to the EAT-Lancet diet.**

|  | Present diet | EAT-Lancet diet |
|---|---|---|
| *Green vegetables* | 50–100 gram/day<br>(at least 5x per week) | 200–600 grams/day |
| *Beef* | 50–60 gram/day<br>(3x /week) | 0–28 grams/day<br>(equals 0–196 grams/week) |
| *Whole milk* | 200–300 ml/day | 0–500 ml/day |
| *Whole dairy butter* | 5 grams/slice of bread<br>(equals 10–20 gram/day) | 0–12 grams/day |

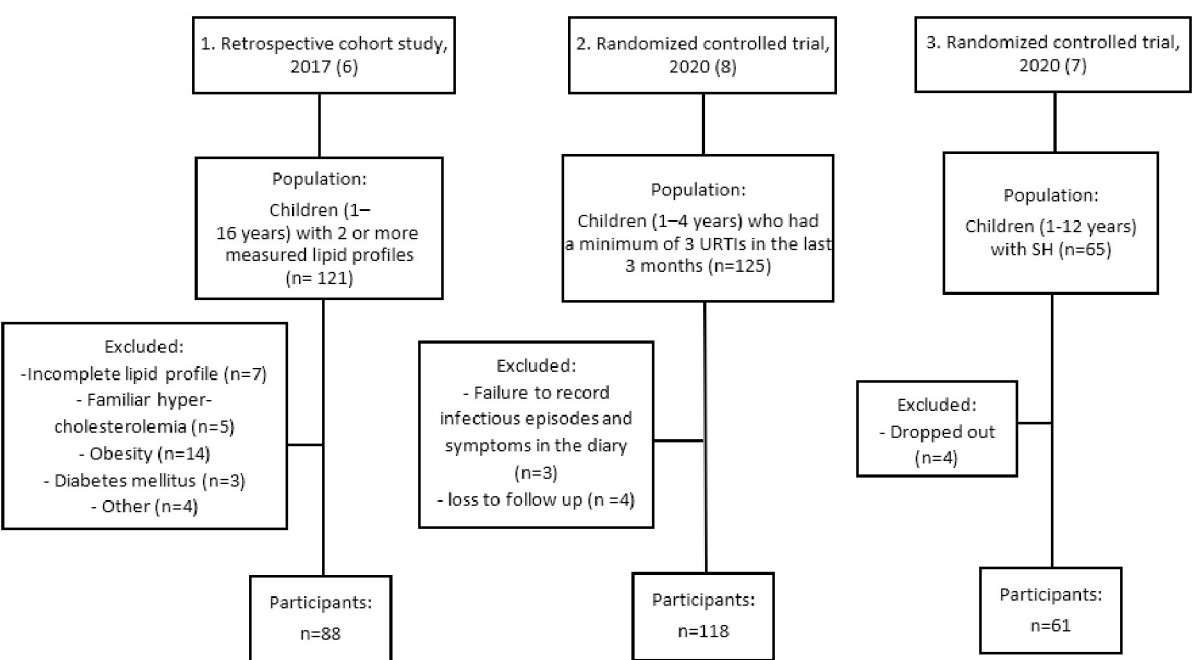

**Fig 1. Schematic overview of the data collection of three previous conducted studies used for this pooled analysis.** One retrospective cohort study and two RCTs.

milk (3.4% fat) and whole dairy butter (80% fat) on each slice of bread. The advice was provided with age-specific portion sizes according to the national guidelines. The control group was not informed about the contents of the dietary advice and continued their usual dietary habits. The primary outcome measurements were the measurements of the lipid profile. Standard Deviation (SD) BMI scores of the children were the secondary outcome measurements. Since absolute BMI scores can't be compared between different age groups in young children, SD scores were determined. BMI was calculated by dividing weight in kilograms by the square of height in meters. The SD BMI score was calculated on the basis of gender, age, height, and weight.

Blood was collected by venapuncture at the start and follow-up of the study. Measurements from the lipid profile are total cholesterol, triglycerides (TG), high-density lipoprotein (HDL), low-density lipoprotein (LDL) cholesterol and total cholesterol/HDL ratio. These measurements of the lipid profile were measured using enzymatic colorimetric techniques with the COBAS 6000 (Roche Diagnostics, Almere, The Netherlands). LDL cholesterol was calculated with Friedewald's formula: LDL = total cholesterol − HDL − (0.45 × TG). SD BMI was calculated based on gender, age, height, and weight. SD scores were used to compare the values with the reference population of Dutch children.

IBM SPSS Statistics 26 (SPSS Inc., Chicago, IL, USA) was used to perform statistical analyses. Raw data of the three intervention studies where combined into one single dataset for analysis. Descriptive statistics were used to obtain mean values with SD, medians, 95% CI and interquartile ranges (IQR). Since this study used results from three previous conducted studies, a mixed effects analysis of covariance (ANCOVA) model was used to test changes in the lipid profile and SD BMI levels between the two groups over time (at intake and at follow-up). Firstly, a one-way analysis of variance (ANOVA) test was performed for every measurement within both the intervention group and control group. Thereafter the ANCOVA test was performed. P-values were considered significant when $<0.05$.

## 3. Results

### 3.1 Baseline characteristics

The non-randomized retrospective cohort study excluded 33 participants. Seven participants had an incomplete lipid profile, 22 participants suffered from a disorder that might influence the lipid profile such as obesity and diabetes mellitus and four participants had other reasons for exclusion. From the first randomized controlled trial, three participants dropped out. Four participants were excluded due to loss of follow-up. Four participants dropped out of the second randomized controlled trial. In total, 267 participants were included in this pooled analysis study. Table 2 shows the descriptive factors age and gender distribution and baseline measurements of the lipid profile of the intervention group and control group per included study of this pooled analysis as well as combined for this pooled analysis. For all dietary components, the compliance was significantly higher in the intervention group. The overall compliance with the dietary advice in the intervention group was 87% compared to 36% in the control group.

### 3.2 Lipid profile

The measurements of the lipid profile at follow-up are shown in Table 3. Within the intervention group, HDL-cholesterol increased significantly from 1.22 mmol/L, 95% confidence interval (CI) 1.14–1.32, to 1.42 mmol/L 95% confidence interval (CI) 1.30–1.65 (p = 0.007). This increase over time was significantly different compared to the increase in the control group from 1.26 mmol, 95% CI 1.19–1.35, to 1.30, 95% CI 1.26–1.37 (p = 0.04). Since HDL-cholesterol increased significantly in the intervention group, the total cholesterol/HDL cholesterol ratio was significantly reduced with 0.45 mmol/L, 95% CI 2.96–3.31 (p = 0.00). This reduction was significantly different (p = 0.05) compared to the reduction of 0.25 mmol/L in the control group 95% CI 3.24–3.64 (p = 0.11). TG was significantly reduced in the intervention group with 0.18 mmol/L, 95% CI 0.99–1.26 (p = 0.001), and in the control group with 0.23 mmol/L, 95% CI 0.99–1.23 (p = 0.03). This reduction was similar between the two groups over time (p = 0.83). Total cholesterol levels and LDL cholesterol did not change significantly within the groups. No significant changes occurred in SD BMI in the intervention group. However, SD

**Table 2. Baseline characteristics and measurements of the lipid profile of the intervention group and control group of the three intervention study's and combined for the pooled analysis.**

| Variable | Intervention group | | | | Control group | | | |
|---|---|---|---|---|---|---|---|---|
| | *1.Retrospective cohort study n = 48* | *2.RCT: URTI's n = 58* | *3.RCT: SH n = 29* | *Pooled analysis n = 135* | *1.Retrospective cohort study n = 40* | *2.RCT: URTI's n = 60* | *3.RCT: SH n = 32* | *Pooled analysis n = 132* |
| *Age in years (median, IQR)* | 2.6 (1.6–8.0) | 2.02 (2.0–2.0) | 7.7 (7.4–4.4) | 4.2 (3.0–4.0) | 4.7 (2.3–9.0) | 1.9 (2.0–1.0) | 8.1 (7.9–4.8) | 4.7 (3.0–5.0) |
| *Male: female (n)* | 25: 23 | 37: 21 | 15: 14 | 77: 58 | 24: 16 | 24: 36 | 16: 16 | 67: 72 |
| *Total cholesterol (mmol/L) ± SD* | 4.34 ± 1.01 | 3.99 ± 0.69 | 4.23 ± 0.99 | 4.12 ± 0.84 | 4.19 ± 0.91 | 3.91 ± 0.81 | 4.29 ± 0.77 | 4.17 ± 0.76 |
| *TG (mmol/L) ± SD* | 1.38 ± 0.74 | 1.46 ±0.80 | 1.14 ± 0.78 | 1.32 ± 0.73 | 1.35 ± 0.94 | 1.44 ± 0.87 | 1.15 ± 0.98 | 1.36 ± 0.92 |
| *HDL cholesterol (mmol/L) ± SD* | 1.20 ± 0.45 | 1.11 ± 0.35 | 1.40 ± 0.46 | 1.22 ± 0.41 | 1.22 ± 0.40 | 1.15 ± 0.36 | 1.47 ± 0.39 | 1.26 ± 0.41 |
| *LDL cholesterol (mmol/L) ± SD* | 2.47 ± 0.98 | 2.22 ± 0.62 | 2.31 ± 0.97 | 2.31 ± 0.79 | 2.32 ± 0.80 | 2.27 ± 0.67 | 2.30 ± 0.74 | 2.32 ± 0.70 |
| *Total cholesterol/HDL cholesterol ratio (mmol/L) ± SD* | 3.96 ± 1.30 | 3.95 ± 1.29 | 3.21 ± 1.39 | 3.70 ± 1.19 | 3.72 ± 1.26 | 3.83 ± 1.58 | 3.19 ± 1.25 | 3.69 ± 1.43 |
| *SD BMI ± SD* | -0.26 ± 1.32 | -0.13 ± 1.16 | 0.36 ± 1.60 | -0.048 ± 1.37 | -0.58 ± 1.21 | -0.061 ± 1.26 | 0.82 ± 1.39 | 0.023 ± 1.38 |

**Table 3. Changes between the two groups in measurements of the lipid profile and SD BMI at the end (follow-up) of the study period (with 95% CI).**

| Measurement | Intervention group (n = 135) | Control group (n = 132) | p-value |
|---|---|---|---|
| Total cholesterol (mmol/L) ± SE | 4.20 ± 0.062 (4.05–4.33) | 4.15 ± 0.064 (4.01–4.29) | 0.34 |
| TG (mmol/L) ± SE | 1.14 ± 0.055 (0.99–1.26) | 1.14 ± 0.059 (0.99–1.23) | 0.83 |
| HDL cholesterol (mmol/L) ± SE | 1.42 ± 0.067 (1.30–1.65) | 1.30 ± 0.034 (1.26–1.37) | 0.04* |
| LDL cholesterol (mmol/L) ± SE | 2.36 ± 0.056 (2.19–2.45) | 2.35 ± 0.056 (2.24–2.48) | 0.86 |
| Total cholesterol/HDL cholesterol ratio (mmol/L) ± SE | 3.25 ± 0.077 (2.96–3.31) | 3.44 ± 0.091 (3.24–3.64) | 0.05* |
| SD BMI ± SE | -0.067 ± 0.13 (-0.42–0.14) | 0.18 ± 0.12 (-0.086–0.43) | 0.05* |

* indicates a significant p-value when p<0.05. SE (Standard Error).

BMI did increase significantly from -0.0054 to 0.18 in the control group 95% CI 0.086–0.43 (p = 0.01). This change over time was significantly different (p = 0.05) compared to the intervention group, which showed a small reduction from -0.045 to -0.06795% CI -0.42–0.14 (p = 0.87).

## 4. Discussion

This pooled analysis aimed to get more information about the effects of a dietary intervention with minimal and unprocessed foods that are relatively high in natural saturated fats, on the lipid profile and SD BMI of young children. There were some concerns about negative effects on the lipid profile regarding the saturated fats. However, consumption of whole dairy products, green vegetables and beef did not result in a negative lipid profile. It even showed a significant increase in the beneficial HDL cholesterol in the children who consumed the dietary advice. This is favourable since higher HDL cholesterol levels can reduce the risk of developing obesity and CVD in later life [9,10].

A previously conducted meta-analysis of randomized control trials in adults supports this finding since it was observed that consumption of natural saturated fatty acids increased HDL cholesterol even more compared to consumption of unsaturated fatty acids [13]. Secondly, a large meta-analysis of prospective cohort studies that investigated the relationship between consumption of fat from whole cow's milk and adiposity in children (aged 1–18) concluded that a higher intake of fat from whole cow's milk was associated with lower adiposity in the children [14]. Furthermore, it had a favourable effect on the lipid profile since TG was reduced and HDL cholesterol was increased. Similar results were observed by Engel et al. (2018) in a 3-week randomized crossover study where whole milk consumption was compared to skimmed milk consumption [15]. Whole milk consumption increased HDL cholesterol significantly compared to skimmed milk. Still, consuming a diet with a high saturated fat percentage is not accepted by many health organizations like the national Heart Foundation of Australia [16]. Besides saturated fats, whole dairy products also contain natural trans-fats, trans-palmitoleic acid, which have been associated with lower LDL cholesterol and TG levels, and higher HDL cholesterol in several large cohort studies [17–19]. However, the mechanism remains unclear. One of the possible mechanisms, by which whole cow's milk does not cause unfavourable lipid profiles, is the presence of calcium. Whole dairy products contain a high amount of calcium, of which several studies observed a positive association with body fat modulation [20–22]. These studies have shown that calcium potentially inhibits fat absorption as faecal fat excretion was increased. A second potential mechanism of saturated fats from whole dairy products is that it reduces chronic inflammation and oxidative stress [23]. Fats from whole dairy products, especially butter, are sources of butyric acid which has anti-inflammatory properties [24]. Butyric acid downregulated the NFκB-mediated inflammatory pathways

resulting in decreased chronic inflammation in the gastrointestinal tract, leading to beneficial effects on the lipid profile, body weight and metabolic health [25,26]. A third possible mechanism that could be involved in lowering measurements of the lipid profile are sphingolipids present in whole dairy products. Ohlsson et al. observed reduced cholesterol absorption in healthy adult men who consumed whole dairy products with sphingolipids [27]. However, they remained inconclusive about the exact mechanism behind this.

The significant increase in HDL cholesterol contributes to the significant decrease in total cholesterol/HDL cholesterol ratio in the intervention group since total cholesterol remained similar after consumption of the diet. The total cholesterol/HDL cholesterol ratio is used as a predictor of developing CVD in later life [1]. Contrary to expectations, this predictor was also slightly decreased in the control group, indicating that not only this dietary advice decreases the risk of developing CVD in later life, but other factors can also influence this measurement. It is known that some commonly consumed food products have lipid-lowering properties. For example, thylakoids (the photosynthetic parts of chloroplasts) and phytochemicals in green vegetables have been shown to lower blood lipids [28]. An intervention study observed significantly lower free fatty acids in serum when healthy normal-weight individuals consumed a high-fat meal with the addition of thylakoids [29]. Secondly, a study with rats treated with thylakoids and phytochemicals from spinach showed a 62.3% reduction in TG [30]. This is in line with results from the present study since the TG were reduced in both groups. However, as expected the compliance to the green vegetables was higher in the intervention group (82%) compared to the control group (57%). Still, this resulted in a similar decrease in TG in both groups, showing green vegetables is not solely responsible for lowering the TG.

The diet used in this intervention consisted of minimal and unprocessed food products, such as green vegetables and whole dairy products. A previous study conducted by Lee et al. (2014) investigated the association between added sugars in (ultra)processed foods and HDL cholesterol in young females in a 10-year follow-up (from 9 years to 19 years of age) cohort study [31]. They observed an increased HDL cholesterol in individuals who consumed less than 10% energy from added sugars. Secondly, previous studies have demonstrated that consumption of (ultra)processed foods in early life contribute to a negative lipid profile in young children [5,32]. The intervention group from this pooled analysis study consumed whole dairy products instead of ultra-processed artificially flavoured dairy products. This could have contributed to the increase in HDL cholesterol [14]. Data of the dietary intake before they started with the dietary advice were incomplete and therefore not shown.

Besides the lipid profile, consumption of whole dairy products, green vegetables and beef did not lead to a change in SD BMI levels in the young children. In contrast, the SD BMI of the children that continued their usual dietary habits increased significantly. This could also be explained by the type of food these children consumed, as they continued their usual dietary habits. UNICEF concluded that 70% of the products made for young children in the Dutch supermarket, such as breakfast cereals and snacks, are unhealthy [3]. These products are consumed by a lot of children in the Netherlands since parents simply do not know that these are unhealthy. According to previous studies, the consumption of (ultra)processed foods leads to poor diet quality [4,33]. Costa et al. (2019) concluded that early (ultra)processed food consumption was associated with increased abdominal obesity in children [34]. Consumption of these products results in negative health outcomes later in life, such as obesity and CVD [3].

It is becoming clear that a healthy and natural diet is necessary to nurture human health and to support environmental sustainability [12]. For children with developing and growing bodies, an adequate intake of macro- and micronutrients is important and required. Food from animal sources can be a valuable source since they contain a high nutrient density [35].

The strength of this pooled analysis is that it combines the results of intervention studies and not of cross-sectional studies. This is not/seldomly done in children of this age group. The number of investigated children is increased compared to each single study and therefore, conclusions can be drawn. Some limitations must be considered as well. The laboratory could not make a distinction in the LDL cholesterol subclasses (large, intermediate or small dense). Previous research has suggested that the LDL subclass pattern and size of the LDL particles are important in the development of CVD [36]. Since this distinction could not be made, LDL cholesterol was categorized as one group and not further analysed. The dietary habits of the children before they started with the dietary advice were incomplete. Suggestions for follow-up studies could be the inclusion of additional information on the composition of the diet (macronutrient composition, fatty acid composition, fiber, plant sterols/stanols) as well as beverages intake. Certainly, children from the investigated age group consume many beverages and notably the caloric ones can influence the lipid profile as well. Secondly, the level of physical activity should be reported in future studies to strengthen the results.

The findings from this pooled analysis can further support the available scientific evidence for nutritional guidelines worldwide. An easily accessible, simple and low-cost diet can lead to beneficial outcomes for young children.

## 5. Conclusion

This pooled analysis, including minimal and unprocessed foods, shows that consumption of whole dairy products, green vegetables and beef had no negative effects on the lipid profile, but increased the prognostic favourable HDL cholesterol significantly. Secondly, this diet had no adverse effect on the SD BMI levels of the children. Therefore, this dietary advice of minimal and unprocessed foods, conform guidelines of the EAT-Lancet sustainable diet for the future, can safely be recommended for children.

## Supporting information

**S1 Data.**
(XLSX)

**S2 Data.**
(XLSX)

## Acknowledgments

The authors would like to thank Professor J. van der Palen (epidemiologist) and Aleksander Peric with their help on the statistical analysis.

## Author Contributions

**Formal analysis:** Rosanne Barbra Hendriksen.

**Investigation:** Rosanne Barbra Hendriksen.

**Methodology:** Ellen José van der Gaag.

**Supervision:** Ellen José van der Gaag.

**Writing – original draft:** Rosanne Barbra Hendriksen.

**Writing – review & editing:** Ellen José van der Gaag.

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
