## [Decision Letter · Decision Letter 0]

23 Apr 2021

PONE-D-21-01126

Effect of a dietary intervention including minimal and unprocessed foods, high in natural saturated fats, on the lipid profile of children

PLOS ONE

Dear Dr. Hendriksen,

Thank you for submitting your manuscript to PLOS ONE. After careful consideration, we feel that it has merit but does not fully meet PLOS ONE’s publication criteria as it currently stands. Therefore, we invite you to submit a revised version of the manuscript that addresses the points raised during the review process.

The manuscript has been evaluated by two reviewers, and their comments are available below.

The reviewers have raised a number of major concerns. Specifically, they request significant improvements to the reporting of methodological aspects of the study. In particular, the authors suggest further refinement to the analytical methods and considerations of further statistical tests.

Could you please carefully revise the manuscript to address all comments raised?

We look forward to receiving your revised manuscript.

Kind regards,

Avanti Dey, PhD

Staff Editor

PLOS ONE

Journal Requirements:

2. For more information on PLOS ONE's expectations for statistical reporting, please see https://journals.plos.org/plosone/s/submission-guidelines.#loc-statistical-reporting. Please update your Methods and Results sections accordingly.

'NO, The funders had no role in study design, data collection and analysis, decision to publish, or preparation of the manuscript'

5. Please amend either the title on the online submission form (via Edit Submission) or the title in the manuscript so that they are identical.

6. Please include a separate caption for each figure in your manuscript.

Reviewers' comments:

Reviewer's Responses to Questions

**Comments to the Author**

1. Is the manuscript technically sound, and do the data support the conclusions?

Reviewer #1: Yes

Reviewer #2: Partly

2. Has the statistical analysis been performed appropriately and rigorously? 

Reviewer #1: No

Reviewer #2: No

3. Have the authors made all data underlying the findings in their manuscript fully available?

Reviewer #1: No

Reviewer #2: No

4. Is the manuscript presented in an intelligible fashion and written in standard English?

Reviewer #1: Yes

Reviewer #2: Yes

5. Review Comments to the Author

Reviewer #1: Specific comments

1) Additional information on the composition of the diet (specific foods, macronutrient composition, fatty acid composition, fiber, plant sterols/stanols) that control subjects consumed as well as the level of physical activity would be useful, not only for the present report but for comparison with previous and future studies.

2) Additional information on the foods (macronutrient composition, fatty acid composition, fiber, plant sterols/stanols) consumed by subjects in the intervention group besides the 4 foods listed in Table 1 as well as the level of physical activity would also strengthen the manuscript.

3) The manuscript indicates that normality was checked by the Shapiro-Wilk test, but was normality found? If not, what action was taken?

4) The manuscript indicates that “All of these external factors did not affect the lipid profile of the children significantly (data not shown)”. The results of these statistical tests should be included in the manuscript whether statistically significant or not.

5) The type and amounts of beverages, notably the caloric ones, consumed by the intervention and control subjects should be documented in the manuscript if possible, and discussed.

Reviewer #2: Title page: the short title 'a pooled analysis' is insufficiently descriptive of this study; please suggest a more informative title. I would also suggest that the main title include the description that this is a pooled analysis e.g. by adding " - pooled evidence from randomised controlled trials and a cohort study".

Abstract: The methods need a bit more detail e.g. one sentence to explain how you conducted the pooled analysis, since this is a key feature of this study. The results in the abstract also don't seem to make sense: for example, you say BMI increased from 0.023 to 0.15 - these are impossibly low values of BMI, unless you meant that these were the changes in BMI (but even then the sentence would still not make sense). Please recheck these. Lastly, the appropriate way to report results is to report the effect/difference, its 95% confidence intervals and the p-value for the hypothesis test of 'no-difference'.

Methods: the current methods are not sufficiently detailed. The study setting, participants, interventions and outcomes are reasonably well reported, but the statistical methods are glossed over. You could add a bit more description, perhaps one or two sentences each, of the underlying studies, e.g. what their objectives were. For the methods, there are several things you need to add to improve the reporting. First, you need to report how you summarised the participant characteristics. This is typically reporting counts and proportions for categorical variables and means or medians with standard deviations or IQRs, respectively, for continuous ones. This should preferably be done separately for each data source (e.g. see Table 1 in http://www.thelancet.com/journals/langlo/article/PIIS2214-109X(17)30484-9/fulltext). Secondly, for this analysis of a pooled dataset you can't simply resort to t-tests, there are statistical issues relating to within-study clustering of the contributing studies to deal with. You need an appropriate method to pool estimates from studies, and this could be done either through (1) analysing the datasets independently (e.g. analysis of covariance/baseline-adjusted models for the continuous outcomes) and then combining the study specific estimates (separately for each outcome) using an appropriate method such as inverse variance weighting to pool them, or (2) combining all the studies into a single dataset and then using a mixed effects ANCOVA/baseline adjusted model on the combined dataset for each outcome, with random effects across studies. Method (1) would be further enhanced by including a forest plot of the study-specific effects and the pooled effect similar to what would be done in a meta-analysis. Comparison of baseline characteristics between treatment groups is unnecessary, especially for randomised trials where baseline imbalance is not expected anyway.

Results: see comment above about comparing baseline characteristics between groups. Furthermore, in table 2 you have also compared them in terms of dietary compliance, which is a feature of the intervention which is necessarily expected to be different - please remove the p-value column from this table. In Table 3 you should not even be including the baseline values in the comparison; as suggested above, you should use the baseline values only for adjustment in the ANCOVA model i.e. a model of the follow-up values with treatment group as the main explanatory variable and including the baseline value of the outcome as a covariate. Table 3 should then report the follow-up mean and standard error (not SD unlike the baseline table) for each arm, followed by the effect (i.e. the coefficient for treatment in the ANCOVA model described above) with its 95% confidence intervals and p-value. You can have additional columns for effect/95%CI/p-value if you report both unadjusted and adjusted effects.

I am unable to assess the manuscript further given the shortcomings of the approach to analysis.

6. PLOS authors have the option to publish the peer review history of their article (what does this mean?). If published, this will include your full peer review and any attached files.

Reviewer #1: No

Reviewer #2: No

---

## [Author Response · Author response to Decision Letter 0]

4 Jul 2021

Dear reviewers,

I would like to thank you for your effort and time you've put into my manuscript. Your input definitely helped to improve the manuscript. I hope you feel the same!

---

## [Decision Letter · Decision Letter 1]

27 Jul 2021

PONE-D-21-01126R1

Effect of a dietary intervention including minimal and unprocessed foods, high in natural saturated fats, on the lipid profile of children, pooled evidence from randomized controlled trials and a cohort study.

PLOS ONE

Dear Dr. Hendriksen,

Thank you for submitting your manuscript to PLOS ONE. After careful consideration, we feel that it has merit but does not fully meet PLOS ONE’s publication criteria as it currently stands. Therefore, we invite you to submit a revised version of the manuscript that addresses the points raised during the review process.

We look forward to receiving your revised manuscript.

Kind regards,

Massimiliano Ruscica, Ph.D.

Academic Editor

PLOS ONE

Reviewers' comments:

Reviewer's Responses to Questions

**Comments to the Author**

1. If the authors have adequately addressed your comments raised in a previous round of review and you feel that this manuscript is now acceptable for publication, you may indicate that here to bypass the “Comments to the Author” section, enter your conflict of interest statement in the “Confidential to Editor” section, and submit your "Accept" recommendation.

Reviewer #1: All comments have been addressed

Reviewer #2: (No Response)

2. Is the manuscript technically sound, and do the data support the conclusions?

Reviewer #1: (No Response)

Reviewer #2: Partly

3. Has the statistical analysis been performed appropriately and rigorously? 

Reviewer #1: (No Response)

Reviewer #2: No

4. Have the authors made all data underlying the findings in their manuscript fully available?

Reviewer #1: (No Response)

Reviewer #2: No

5. Is the manuscript presented in an intelligible fashion and written in standard English?

Reviewer #1: (No Response)

Reviewer #2: Yes

6. Review Comments to the Author

Reviewer #1: (No Response)

Reviewer #2: Abstract

- SD BMI is introduced suddenly in the abstract without explanation; this should be described. Is it the standardised body mass index? If so, say so and also explain how the standardisation was done.

- the methods should summarise how the pooled analysis was conducted; there is nothing in the current abstract that gives the reader any sense of what this pooled analysis constituted: for example, did you combine all data into a single dataset and then conduct regressions looking at the associations between exposures and outcomes in this combined/pooled dataset? If so, a sentence saying exactly this should be included in the methods after the sentence that currently ends on line 52.

- the statements from the second half of line 46 to the end of line 47 pertaining to the numbers included in the analysis are results and should be moved to the results section of the abstract.

- I appreciate that the authors have included effect estimates, confidence intervals and p-values in the results. However there is a problem with the results as presented. The point estimate of the difference between intervention and control group of 0.20 would be expected to lie within the confidence intervals; so a confidence interval of 1.30 to 1.65 is not consistent with this point estimate. Additionally, the correct order to cite the numerical results is: estimate 95%CI p-value, not estimate p-value 95%CI as currently done on line 55-56. Lastly, the result at the end of line 56-57 should also include an estimate of the magnitude of increase with 95%CI and p-value. I do wonder though what the relevance is of reporting changes over time in the control group - it is the difference in difference over time between the intervention and control groups which would be important to report here.

Methods

- more clarity is needed on what SD BMI means, especially how the standardisation was done to obtain this outcome measure.

- more clarity is needed on how the 'pooling' was done here: did you pool raw data from the studies into a single dataset then analyse them, or did you pool the estimates from the three studies to obtain pooled estimates?

- in line 129 you say participants were divided into control group and intervention group without describing what those groups received. This statement as it serves no particular purpose (and you say nothing about the same for the other two studies), and you should probably remove it from this section, as you later describe the interventions from line 141.

- I think you should remove all statements about tests for normality; they are not useful, as they are extremely sensitive to very slight deviations from normality. In any case, normality of outcomes in not a requirement for regression analyses or ANOVA/ANCOVA (multivariate normality or normality of residuals is what is important). Furthermore, the statements about outcome measures in line 156 to 158 are results and would not be included in the methods even if they were relevant. Please remove these and the tests for normality.

Results

- effects or differences should be reported in this format: effect or difference, 95% confidence interval, p-value.

- as noted in the abstract, it is not possible for the increase of 0.20 mmol/L to have a confidence interval that does not straddle/include 0.20; the lower and upper 95%CI of 1.30 to 1.65 do not include 0.20 so either the point estimate or the confidence interval is incorrect.

- it would be more meaningful to present the difference in pre-treatment and post-treatment values of the outcomes compared between the intervention and controls groups i.e. difference-in-difference analysis. To do this, table 1 should include the baseline means and SDs of each of the outcomes in each group. Table 2 should then include the mean and SE (not SD) of each outcome in each arm in the follow-up period, along with the baseline-adjusted differences between the groups with 95%CI and p-values. You may also further adjust for other relevant factors if desired.

Please consult a statistician to advice on these and other issues with the analysis and presentation of results.

7. PLOS authors have the option to publish the peer review history of their article (what does this mean?). If published, this will include your full peer review and any attached files.

Reviewer #1: No

Reviewer #2: No

---

## [Author Response · Author response to Decision Letter 1]

29 Sep 2021

We would like to thank the reviewer for the time and valuable comments on the manuscript. All comments were carefully studied and answered. The comments were valuable for the improvements of this manuscript.

---

## [Decision Letter · Decision Letter 2]

3 Dec 2021

Effect of a dietary intervention including minimal and unprocessed foods, high in natural saturated fats, on the lipid profile of children, pooled evidence from randomized controlled trials and a cohort study.

PONE-D-21-01126R2

Dear Dr. Hendriksen,

We’re pleased to inform you that your manuscript has been judged scientifically suitable for publication and will be formally accepted for publication once it meets all outstanding technical requirements.

Kind regards,

Massimiliano Ruscica, Ph.D.

Academic Editor

PLOS ONE

Additional Editor Comments (optional):

Reviewers' comments:

Reviewer's Responses to Questions

**Comments to the Author**

1. If the authors have adequately addressed your comments raised in a previous round of review and you feel that this manuscript is now acceptable for publication, you may indicate that here to bypass the “Comments to the Author” section, enter your conflict of interest statement in the “Confidential to Editor” section, and submit your "Accept" recommendation.

Reviewer #2: All comments have been addressed

2. Is the manuscript technically sound, and do the data support the conclusions?

Reviewer #2: (No Response)

3. Has the statistical analysis been performed appropriately and rigorously? 

Reviewer #2: (No Response)

4. Have the authors made all data underlying the findings in their manuscript fully available?

Reviewer #2: (No Response)

5. Is the manuscript presented in an intelligible fashion and written in standard English?

Reviewer #2: (No Response)

6. Review Comments to the Author

Reviewer #2: (No Response)

7. PLOS authors have the option to publish the peer review history of their article (what does this mean?). If published, this will include your full peer review and any attached files.

Reviewer #2: No

---

## [Editor Report · Acceptance letter]

14 Dec 2021

PONE-D-21-01126R2 

Effect of a dietary intervention including minimal and unprocessed foods, high in natural saturated fats, on the lipid profile of children, pooled evidence from randomized controlled trials and a cohort study. 

Dear Dr. Hendriksen:

I'm pleased to inform you that your manuscript has been deemed suitable for publication in PLOS ONE. Congratulations! Your manuscript is now with our production department. 

Kind regards, 

on behalf of

Dr. Massimiliano Ruscica 

Academic Editor

PLOS ONE